Rapid and cost-effective nutrient content analysis of cotton leaves using near-infrared spectroscopy (NIRS)

Prananto Jeremy Aditya 1 jayyprananto9@gmail.com
Minasny Budiman 1
Weaver Timothy 2
1 School of Life and Environmental Sciences, The University of Sydney, Sydney Institute of Agriculture , Sydney, NSW , Australia
2 CSIRO Agriculture and Food , Myall Vale, NSW , Australia
Adhikari Kabindra
Electronic publication date: 2021 Mar 11
Publication date: 2021
Volume: 9
Electronic Location ID: e11042
Received 2020 Aug 24; Accepted 2021 Feb 9
Copyright: © 2021 Prananto et al.
Copyright year: 2021
Copyright holder: Prananto et al.
License: This is an open access article distributed under the terms of the Creative Commons Attribution License, which permits unrestricted use, distribution, reproduction and adaptation in any medium and for any purpose provided that it is properly attributed. For attribution, the original author(s), title, publication source (PeerJ) and either DOI or URL of the article must be cited.
License URL: https://creativecommons.org/licenses/by/4.0/

Keywords: Plant nutrient management, Near-infrared spectroscopy (NIRS), Plant nutrient analysis, Proximal sensing, Phenotyping, Cotton nutrient management, Portable spectrometers, Real-time sensing

Funding: Sydney Institute of Agriculture CSIRO Agriculture & Food Strategic Investment Project (SIP) Funds This work was supported by the Sydney Institute of Agriculture on Nitrogen use efficiency and CSIRO Agriculture and Food Strategic Investment project (SIP) funds. The funders had no role in study design, data collection and analysis, decision to publish, or preparation of the manuscript.

==============================
The development of portable near-infrared spectroscopy (NIRS) combined with smartphone cloud-based chemometrics has increased the power of these devices to provide real-time in-situ crop nutrient analysis. This capability provides the opportunity to address nutrient deficiencies early to optimise yield. The agriculture sector currently relies on results delivered via laboratory analysis. This involves the collection and preparation of leaf or soil samples during the growing season that are time-consuming and costly. This delays farmers from addressing deficiencies by several weeks which impacts yield potential; hence, requires a faster solution. This study evaluated the feasibility of using NIRS in estimating different macro- and micronutrients in cotton leaf tissues, assessing the accuracy of a portable handheld NIR spectrometer (wavelength range of 1,350–2,500 nm). This study first evaluated the ability of NIRS to predict leaf nutrient levels using dried and ground cotton leaf samples. The results showed the high accuracy of NIRS in predicting essential macronutrients (0.76 ≤ R2 ≤ 0.98 for N, P, K, Ca, Mg and S) and most micronutrients (0.64 ≤ R2 ≤ 0.81 for Fe, Mn, Cu, Mo, B, Cl and Na). The results showed that the handheld NIR spectrometer is a practical option to accurately measure leaf nutrient concentrations. This research then assessed the possibility of applying NIRS on fresh leaves for potential in-field applications. NIRS was more accurate in estimating cotton leaf nutrients when applied on dried and ground leaf samples. However, the application of NIRS on fresh leaves was still quite accurate. Using fresh leaves, the prediction accuracy was reduced by 19% for macronutrients and 11% for micronutrients, compared to dried and ground samples. This study provides further evidence on the efficacy of using NIRS for field estimations of cotton nutrients in combination with a nutrient decision support tool, with an accuracy of 87.3% for macronutrients and 86.6% for micronutrients. This application would allow farmers to manage nutrients proactively to avoid yield penalties or environmental impacts.

Introduction

Cotton production in Australia is one of the most important agricultural industries in terms of export. Approximately 99% of Australia’s cotton is exported, contributing two billion dollars towards the Australian economy in 2018 (Cotton Australia, 2018). One of the most challenging issues currently impacting the industry is inefficient nutrient management (Beegle, Carton & Bailey, 2000; Dobermann, 2007; Ali et al., 2008). For instance, the timing of nitrogen (N) application at crucial physiological stages determines the crop development, yield potential and fibre quality (Read, Reddy & Jenkins, 2006; Rochester, 2011; Yang et al., 2011). However, farmers and their consultants tend to frequently over-supply N to avoid a yield penalty from N deficiency. Oversupplying N can also have a negative impact on cotton, that is, environmental through nitrous oxide emissions and economical (Deutscher, Bange & Rochester, 2001; Rochester, 2011). Thus, a more strategic management of N and other nutrients is desirable to improve the overall yield, quality, and income of cotton production, while reducing negative environmental impacts.

Efficient crop nutrient management relies on the ability to balance the nutrient demand of crops with the supply from the soil (Van Noordwijk & Cadisch, 2002; Fageria & Baligar, 2005; Ulissi et al., 2011). It is important to note that this crop nutrient demand is dynamic, as it changes in terms of both quantity and quality throughout the season (Hallikeri et al., 2010). Ensuring adequate nutrients are available in an accessible form during key stages of crop growth is crucial in determining the overall yield and fibre quality in cotton (CottonInfo, 2021).

Split application is one management strategy that divides nutrient application, for example, 30% before sowing and 70% allocated during crucial physiological periods (Hallikeri et al., 2010). This strategy improves the nutrient use efficiency by allowing nutrients to be adjusted and allocated more efficiently to meet the crop’s dynamic nutrient demand (Hallikeri et al., 2010). To properly execute this split application strategy, the ability to monitor the crop’s nutrient status during the season would be integral. Knowing the nutrient status of the crop in real-time would provide growers with the ability to adjust their nutrient applications at critical stages of cotton crops more accurately and effectively (Tarpley, Reddy & Sassenrath-Cole, 2000; Samborski, Tremblay & Fallon, 2009).

Current methods of monitoring and determining the nutrient status or demand for cotton crops are based on soil and less often on leaf/petiole nutrient analysis. One of the issues with soil analysis is it only provides the overall nutrient content present in the soil (Houba et al., 1996); however, it does not provide information on the plant available nutrients due to different nutrients existing in different forms and availability to the plant. For example, total nitrogen is commonly measured in soil analysis; whereas only a small portion of total nitrogen, mainly nitrate is available for plants (Miley, Maples & Keisling, 1990). Moreover, this nutrient monitoring method requires time-consuming and expensive laboratory analysis. Due to the significant amount of time between sampling and when results are obtained, farmers are forced to be reactive in their nutrient management. Due to the dynamic nature of the nutrients in the soil and plant, especially N (Grundon, 2006), analysis results that are delayed become irrelevant. Thus, for split application in cotton crops to be more effective, a monitoring method that can provide real-time analysis of the nutrient status of crops is desirable.

Handheld near-infrared spectroscopy (NIRS) technology is an emerging technique that can potentially provide a rapid, cheaper, and more efficient solution to the time-consuming and expensive laboratory analysis (García-Martínez et al., 2012; Van Maarschalkerweerd & Husted, 2015). These handheld devices utilise a design that incorporates a miniaturised spectrometer making them more compact. While research-grade portable visible to near infrared (Vis-NIR) instruments have been proven as the standard in measurement, the use of emerging miniaturised spectrometer would offer value to growers and consultants.

The most crucial advantage of NIRS is that it can provide real-time in-situ analysis of the plant nutrient status. This would allow growers to be more proactive and efficient in their nutrient management. Therefore, it would be beneficial to study the potential use of a portable NIR instrument for estimating in-situ plant nutrient status.

Near-infrared spectroscopy has been used extensively in the agriculture industry, including grain quality analysis, dry matter or sugar analysis of fruits (e.g. grapes, avocados, or mangoes), forage quality analysis, and soil analysis (Peng et al., 2015; Tang, Jones & Minasny, 2020; Ng et al., 2019). There are few published results in the literature that have explored the use of NIRS for the nutrient analysis of various plants such as mustards (Martínez-Valdivieso, Font & Río-Celestino, 2019), sorghum, oat, and corn (Savi et al., 2019), wheat and barley (Zerner & Parker, 2019), vine and grape berries (Cuq et al., 2020), and peach (Dedeoglu, 2020). The use of NIRS for the determination of plant nutrient status in cotton is limited and was reported in a study by Tarpley, Reddy & Sassenrath-Cole (2000) in the United States of America. Therefore, it would be essential to conduct a study on the application of NIRS in estimating various macro- and micronutrients in cotton crops, which can potentially be used as a tool that can provide a real-time and accurate nutrient status of cotton crops for nutrient management.

Most studies on utilising NIRS in estimating the nutrient status of leaves were done on dried and ground leaf samples, which requires an extensive amount of labour and time (drying and grinding the sample) (Galvez-Sola et al., 2015). The ability to measure the nutrient status of fresh leaves in real-time and in-field would be highly desirable. Hence, it would be useful to analyse the feasibility of applying NIRS on fresh leaves.

This study evaluated the performance of NIRS in estimating important macro- and micronutrients of cotton leaf tissues using an emerging low-cost handheld NIR spectrometer. Additionally, this study developed a NIRS model from fresh leaves which has the potential to be used in field conditions. The accuracy of these fresh leaf models was compared to models built using dried and ground leaves. Lastly, this study assessed the effectiveness of using NIRS predicted leaf nutrient status information on nutrient decision support tools.

Materials and Methods

Cotton leaf NIR spectra acquisition

NIR reflectance spectra (Fig. 1) were collected from 375 cotton leaf samples sourced from the Australian Cotton Research Institute (ACRI), Myall Vale. The leaves were collected from two recently released Bollgard® three cotton varieties, Sicot 714B3F and Sicot 746B3F.

Figure 1 Reflectance spectra of cotton leaves at different sample preparations.

(A) Reflectance spectra pattern of fresh and intact leaves. (B) Reflectance spectra pattern of fresh and removed leaves. (C) Reflectance spectra pattern of dried and ground leaves that were scanned using the PhoneLab™. The reflectance spectra were comparable between fresh and intact and fresh and removed cotton leaves; however, the reflectance spectra of dried and ground cotton leaves were distinctive from fresh cotton leaves.

The handheld NIR spectrometer (PhoneLab™) (Fig. 2) used in this study comprises a single photodetector, 1,300–2,500 nm wavelength range (NIR range) with a 16 nm resolution manufactured by Si-Ware Systems similar to the Neospectra system used by Tang, Jones & Minasny (2020). The wavelength accuracy is ±1.5 nm with repeatability of ±0.15 nm. The spectrometer is developed based on a semiconductor Micro Electro-Mechanical Systems microfabrication techniques. It uses a 3.3 V power supply, which powers three configurable lamps with a fibre optic light source. The light beam collects at a diameter of 2.5 mm. The spectrometer uses the ceramic reflector with one side glazed 99AL2O3, ODD22*3 mm white reference. This spectrometer is an emerging low-cost (~$2,500) NIR instrument, weigh less than 0.5 kg, with a dimension of 7.5 × 5 × 5 cm.

Figure 2 Picture of the near infrared spectroscopy equipment used in this study.

The system includes the handheld NIR spectrometer (PhoneLab™) and the user friendly software interface that can be used on android devices.

Spectrometers were warmed up for 30 min before use to reduce noise in the spectra. The spectrometers were calibrated using a white reference at the beginning of the scans and after every three scans. All scans were conducted with an aluminium foil background to maintain a consistent depth of penetration. NIR reflectance spectra were taken at three different sample conditions;

Fresh and intact leaves

The first set of NIR reflectance spectra were collected from fresh leaves that were still intact on the plants inside a glasshouse (see Fig. S1A). The glasshouse microclimate mimicked an ideal growing condition of cotton in the field, with a temperature between 30 °C and 35 °C and relative humidity of 60–70%. The NIR scans were conducted on 115 samples, where each sample comprised of two leaves from the same plant. The NIR scans were administered at the base of the leaf close to the main veins at three replicates (different positions) for each leaf, which resulted in a total of six scans per sample. Leaf samples were collected from different parts of the plant to optimise the variation in nutrient content, as some nutrients such as N are quite mobile within the plant (Marschner, 2011).

Fresh and removed leaves

After the first set of scans, the leaves were removed from the plants (see Fig. S1B). The leaves were then left in room condition (25 °C) for 1–2 h to simulate a situation where scanning at the field is not possible, in which an analyst would remove the leaves and scan them in a more controlled environment. The leaves were scanned using the same protocol as the fresh and intact scans with 6 replicates for each sample. For fresh and removed scans, there were an additional 114 samples that were scanned in the previous summer 2019.

Dried and ground leaves and petioles

After the second set of scans, each sample of two leaves was oven-dried at 60 °C until constant weight to ensure moisture was removed from the leaves. The petioles were separated from the leaves. Then, both leaf and petiole samples were ground passed through a 0.25 mm sieve using the FOSS Cyclotec™ 1903 grinder. The dried and ground leaf samples (Fig. S1C) weighing approximately 2.5–5 g depending on the leaf size were spread on the aluminium foil and flattened to a thickness of 2–3 mm for leaf samples and a minimal of 1 mm thickness for petiole samples, fully covering the NIR sensor. The dried and ground leaf and petiole samples were then scanned three times at different positions to ensure a representative reading. For dried and ground samples, there were an additional 114 samples that were scanned in the previous summer 2019 and another 146 samples that were collected from the previous growing season (2017/2018).

Scanning of dried and ground leaf samples started from March 2019, whereas the fresh leaf samples were scanned during winter in July 2019. As it was not possible to scan all leaves under fresh and intact or fresh and removed conditions (due to time constraint and growing season), only 115 cotton leaf samples were scanned under three conditions (dried and ground, fresh and removed, fresh and intact), and only 115 leaf samples had petiole data. The reflectance spectra trend of cotton leaves at different sample preparation can be seen in Fig. 1.

Nutrient analysis of cotton leaf

The dried and ground leaf and petiole samples were analysed in a commercial laboratory (CSBP Plant and Soil Analysis Laboratory) to obtain nutrient data. Nutrients that were assessed included macronutrients (leaf and petiole total Nitrogen (total N), nitrate-nitrogen (nitrate-N), phosphorus (P), potassium (K), sulfur (S), calcium (Ca) and magnesium (Mg)) and micronutrients (molybdenum (Mo), manganese (Mn), zinc (Zn), iron (Fe), sodium (Na), chloride (Cl), boron (B) and copper (Cu)). The methods for each nutrient are listed below:Total nitrogen (total N) was measured using Method 9G2 of Rayment & Lyons (2012), which used the Dumas high temperature combustion method (LECO CN 925 analyser).

Nitrate-N and Cl were extracted using deionised water. Extracted nitrate was reduced to nitrite via copperised cadmium column and measured colourimetrically. The extracted Cl was reacted with mercuric thiocyanate, which released the thiocyanate ions that was reacted with ferric ions and measured colourimetrically.

P, K, Ca, Mg, Cl, Na, Fe, Zn, B, Mn and Cu were extracted using hydrogen peroxide and nitric acid solution and analysed via the ICP (Inductively coupled plasma) spectroscopy (McQuaker, Brown & Kluckner, 1979).

Mo was digested using hydrogen peroxide and nitric acid mixture and the digests were read using ICP-MS (Inductively coupled plasma mass spectroscopy) (McQuaker, Brown & Kluckner, 1979).

The nutrient data of the cotton leaves from the commercial laboratory analysis are shown in Table 1.

Table 1 Summary of the nutrient content values of the cotton leaf samples.

The summary includes the median value and range of the nutrients’ content value of the cotton leaf samples (n = 375) that were used to establish the calibration.

Nutrient			
Macronutrient	Median (%)	Range (%)	
Total nitrogen	3.59	1.53–5.49	
Phosphorus	0.36	0.14–0.87	
Potassium	1.58	0.69–4.23	
Calcium	4.23	2.04–7.26	
Magnesium	0.70	0.37–1.38	
Sulfur	1.00	0.57–3.09	
Petiole total N*	1.99	1.07–4.47	
Nitrate-N	764.06 (mg/kg)	40.00–4,611.38 (mg/kg)	
Micronutrient	Median (mg/kg)	Range (mg/kg)	
Iron	184.27	73.65–900.80	
Manganese	86.21	27.50–367.80	
Copper	6.52	1.92–12.50	
Zinc	19.39	8.14–64.62	
Molybdenum	781.41	153.27–2,937.57	
Boron	99.46	43.15–185.88	
Chloride	1,2300.00	6,700–30,000	
Sodium	1,000.00	200–5,300	
Note:

* Only had 115 samples

Spectra pre-processing

The raw spectra (Fig. 1) were processed prior to the multivariate calibration analysis to remove any significant noise. All spectra processing, modelling, and data analyses were performed using R statistical software. The spectra pre-treatments were kept consistent across all treatments to gain true comparisons of the different treatments (e.g. spectra wavelength/device used and sample treatment). First, the reflectance spectra from each sample (six replicates per sample for fresh leaves, three replicates per sample for dried and ground leaves) were averaged. The averaged reflectance spectra were allocated to standard wavelength positions with 5 nm spacing via spline interpolation using the ‘resample’ function from the prospectr package in R (Stevens & Ramirez-Lopez, 2014). Then, the 5 nm resolution spectra were subjected to a detrending function to remove scattering effect using the ‘detrend’ function in the prospectr package (Stevens & Ramirez-Lopez, 2014), where the spectra were normalised using the SNV (Standard Normal Variate) transformation followed by fitting a second-order polynomial model. The residuals of the trend were utilised as a detrended spectra.

Establishing calibration models

The pre-processed reflectance spectra (“Spectra Pre-Processing”) and the nutrient analysis data (“Nutrient Analysis of Cotton Leaf”) were used to build the calibration models. Models were built for each of the nutrients, and three models were calibrated using leaf reflectance spectra at three sample conditions (i.e. fresh and intact leaves, fresh and removed leaves, and dried and ground leaves).

Calibration models were built using the Cubist regression tree (Kuhn & Johnson, 2013). Cubist is a rule-based model commonly used in analysing NIR spectra data in soil studies (Peng et al., 2015; Tang, Jones & Minasny, 2020; Ng et al., 2019). Cubist creates a tree model by splitting the data based on the selected reflectance wavelength to minimise the error of prediction. Within each split (subset of the data), Cubist fits a linear model using spectra as predictors. This results in a set of ‘if-then’ rules where within each rule, a linear model of spectra model is prescribed (Kuhn & Johnson, 2013). The Cubist regression model was used in this study as it was shown to provide comparable or better calibration compared to the conventional partial least squares regression technique (Tang, Jones & Minasny, 2020, See also Supplemental Information).

The dataset was split randomly into 75% for calibration data and the other 25% was used as validation data to assess the predictive accuracy of the model. Accuracy parameters of the models included: coefficient of determination (R2), Lin’s concordance, root mean square error (RMSE), and bias as described in Nicolai et al. (2007). The random 75:25 data split and calibration processes were repeated 50 times to gain an average measurement of the accuracy parameters.

Statistical analysis

The accuracy of the prediction was assessed using R2 and RMSE. The R2 was calculated using Eq. (1) with np the number of objects used in the calibration or validation, yi¯ the average of al the observation and yi^ and yi the predicted and measured value of the ith observation respectively, shows the proportion of the variance of the calibration or validation dataset explained by the model and RMSE was calculated using Eq. (2) measures the goodness of fit of the model (Nicolai et al., 2007).

(1) R2=1−∑i=1np⁡(yi^−yi)2∑i=1np⁡(yi¯−yi)2

(2) RMSE=∑i=1np⁡(yi^−yi)2np

The statistical distribution of the R2 values from the 50 repetitions of the calibration and validation predictions of the models constructed from the NIR leaf spectra in predicting macro- and micronutrient will be portrayed as a boxplot for comparisons. First, the statistical distribution of the average R2 values of the models calibrated from the NIR (1,350–2,500) in predicting all macro- and micronutrients on the calibration and validation datasets were compared. These models were developed from the 375 dried and ground leaf samples. Dried and ground leaf samples were used as standards because NIR reflectance can be affected by moisture and particle size (Ludwig & Khanna, 2001). In this way, we can assess the accuracy of NIR spectrometer without the effect of external conditions (moisture, temperature, etc.).

The validation accuracy parameter, RMSE from the 50 repetitions, was then used to compare the NIR models using a paired t-test analysis (α = 0.01) to assess the significance of the different leaf treatments. The RMSE was chosen to represent the quality of the calibrations because it portrays the goodness of fit of the calibrations, where a lower RMSE indicates a calibration with higher accuracy.

Finally, important wavelengths used by the Cubist models for the prediction of the different nutrients were extracted to analyse the relationship between the NIR reflectance spectra and the corresponding nutrient content. The three most important wavelengths used in the 50 iterations of the Cubist model to estimate each nutrient were recorded and displayed in a density plot to assess the importance of different regions of the spectra in estimating the assessed nutrients.

Application in nutrient decision support system analysis

NutriLOGIC is a commonly used decision support tool in the Australian cotton industry that can aid in the identification of deficient and excessive levels of various macro- and micronutrients most commonly used for N management (Deutscher, Bange & Rochester, 2001). NutriLOGIC uses an algorithm that categorises the nutrient levels of a cotton plant into three levels: (1) High, (2) Normal and (3) Low based on the developmental stage of the crop and its current nutrient status. This would help farmers decide whether any fertiliser application is required.

A total of 25 samples were randomly selected from ACRI in Myall Vale to test the efficacy of the fresh leaf NIRS models when used in the NutriLOGIC system. The measured and predicted concentrations of each nutrient were inserted into NutriLOGIC, where each nutrient for each sample will be given a category. The resulting analysis of N, P, K, S, Ca, Mg were grouped as macronutrients and Na, Zn, Fe, Cu, Mn, B were grouped as micronutrients. The analysis of both macro- and micronutrients status were compared, and the categorisation error of the predicted nutrient status were assessed to determine whether the accuracy of fresh leaf models is acceptable in practice.

Results

“Nutrient Calibration” compares the calibration and validation prediction accuracy of models calibrated from the NIR spectra of dried and ground leaf samples in predicting leaf macro- and micronutrients. “Macronutrients” and “Micronutrients” detail the performance of the NIRS models in predicting macronutrients and micronutrients, respectively. “Leaf Sample Preparation” compares the prediction accuracy of calibrations constructed from leaves under different sample treatments or conditions. “Effectiveness of Leaf NIRS Models on a Nutrient Decision System” demonstrates the effectiveness of fresh leaf NIRS models in the application of nutrient decision support tools.

Nutrient calibration

Figure 3 shows that the validation R2 values were generally lower and more widely distributed compared to their corresponding calibration R2, which stressed that reporting only calibration statistics can be over-optimistic. Thus, all results in the following sections were based on the validation dataset. Additionally, NIRS models predicted macronutrients (median R2 = 0.85) more accurately compared to micronutrients (median R2 = 0.70).

Figure 3 Boxplot comparing the statistical distribution of the calibration and validation R2 of NIRS models in predicting macro- (A) and micronutrient (B) content in cotton leaves.

Statistical distribution is available in Table S1.

Macronutrients

Overall, all macronutrients were accurately predicted by NIRS, except for nitrate-N which had the lowest validation R2 of 0.41. The complete accuracy parameters of the NIR models in estimating macronutrients are shown in Tables S2. Leaf total N had the highest validation R2 at 0.94 followed by Ca, S and Mg with validation R2 values ≥ 0.80. Validation R2 values for P and K were relatively lower compared to other macronutrients.

Despite the high prediction accuracy of total N, nitrate-N was poorly predicted by NIRS with a low validation R2 of 0.41, and a large bias (Tables S2 and S3). Petiole total N, however, was better predicted compared to nitrate-N by NIRS models. The relatively lower validation accuracy of the petiole total N might be due to the smaller sample size (n = 115) used in the calibration. There is potential for the calibration to be improved further using a larger sample size with a wider and well-distributed petiole total N content.

Micronutrients

Sodium had the highest validation R2 (0.76). Micronutrients that had a higher validation R2 value compared to the R2 median value for micronutrients (0.70) were Mo, Mn, Na, Fe. Validation R2 values for B and Cl were around the median values. Zinc was the least accurately predicted micronutrient with a validation R2 of 0.36 (NIR). Hence, NIRS cannot be used to predict Zn content in cotton leaves. The complete accuracy parameters of the NIRS calibration in predicting different micronutrients are shown in Table S3.

Leaf sample preparation

The following sections show the potential application of the NIR spectrometer in the field by comparing the accuracy of the models developed from cotton leaves at different conditions.

Dried and ground vs. fresh and removed samples

The paired t-test output comparing the prediction accuracy between models built using dried and ground leaf samples and fresh and removed leaf samples in predicting macro- and micronutrients are shown in Table 2. Based on the validation RMSE, macronutrients that were predicted with equal accuracies by both leaf sample conditions were P, K, Mg and nitrate-N. Prediction using models derived from dried and ground leaves were significantly more accurate (p < 0.01) for total N, Ca and S compared to fresh and removed leaf models. Most micronutrients were significantly predicted more accurately using dried and ground leaf models. There were only two micronutrients (Cl and Mn) that were predicted with a similar accuracy by models derived from both leaf conditions. Boron was the only micronutrient that was predicted more accurately by fresh and removed models.

Table 2 Results of the paired t-test between the 75:25 split validation RMSE of the dried and ground cotton leaf and the fresh and removed cotton leaf macronutrient and micronutrient models that were constructed from the NIR spectra captured using the PhoneLab™.

The results include the p-value of the paired t-test. The complete accuracy parameters of the dried and ground and fresh and removed calibrations are available in Table S4.

	Dried and ground	Fresh and removed	p-Value	Preferred sample preparation	
Macronutrient	Validation RMSE (%)	Validation RMSE (%)			
Total nitrogen	0.21	0.29	<0.001	Dried and ground	
Phosphorus	0.09	0.09	0.066	–	
Potassium	0.37	0.37	0.867	–	
Calcium	0.36	0.53	<0.001	Dried and ground	
Magnesium	0.09	0.09	0.179	–	
Sulfur	0.19	0.32	<0.001	Dried and ground	
Petiole total N*	0.38	0.38	0.787	–	
Nitrate-N+	685.52	713.09	0.567	–	
Micronutrient	Validation RMSE (mg/kg)	Validation RMSE (mg/kg)			
Iron	64.2	76.29	<0.001	Dried and ground	
Manganese	44.14	45.59	0.225	–	
Copper	1.10	1.35	<0.001	Dried and ground	
Zinc	8.07	7.02	0.004	Fresh and removed	
Molybdenum	203.98	255.47	<0.001	Dried and ground	
Boron	19.36	17.44	<0.001	Fresh and removed	
Chloride	2300	2700	0.009	–	
Sodium	400	500	<0.001	Dried and ground	
Notes:

* n = 115.

+ RMSE (mg/kg).

Fresh and removed vs. fresh and intact samples

Table 3 shows the paired t-test comparing the prediction accuracy of the models derived from fresh and removed leaves and fresh and intact leaves in predicting macro-and micronutrients. Most macronutrients were predicted with similar accuracies by both the fresh and removed and fresh and intact leaf models (p < 0.01). The majority of micronutrients were predicted with similar accuracies by fresh and removed and fresh and intact leaf models, except for Cu (better under fresh and removed leaves) and Mn and Na (better under fresh and intact).

Table 3 Results of the paired t-test between the 75:25 split validation RMSE of the fresh and removed cotton leaf and fresh and intact cotton leaf macronutrient and micronutrient models that were constructed from the NIR spectra captured using the PhoneLab™.

The results include the p-value of the paired t-test. The complete accuracy parameters of the dried and ground and fresh and removed calibrations are available in Table S5.

	Fresh and removed	Fresh and intact	p-Value	Preferred sample preparation	
Macronutrient	Validation RMSE (%)	Validation RMSE (%)			
Total nitrogen	0.26	0.28	0.035	–	
Phosphorus	0.11	0.11	0.509	–	
Potassium	0.44	0.41	0.028	–	
Calcium	0.63	0.56	0.005	Fresh and intact	
Magnesium	0.11	0.12	0.089	–	
Sulfur	0.38	0.38	0.953	–	
Petiole total N	0.34	0.31	0.015	–	
Nitrate-N+	832.85	772.09	0.116	–	
Micronutrient	Validation RMSE (mg/kg)	Validation RMSE (mg/kg)			
Iron	85.65	68.85	0.013	–	
Manganese	60.62	52.94	<0.001	Fresh and intact	
Copper	1.15	1.26	0.001	Fresh and removed	
Zinc	9.76	9.06	0.186	–	
Molybdenum	176.38	166.22	0.200	–	
Boron	20.08	18.95	0.119	–	
Chloride	2,800.00	2,700.00	0.868	–	
Sodium	281.83	251.16	0.002	Fresh and intact	
Note:

+ RMSE (mg/kg)

Effectiveness of leaf NIRS models on a nutrient decision system

This section compared the analysis results of NutriLOGIC, a decision support system used in the Australian cotton industry for fertiliser recommendations (when supplied with laboratory analysis measurements) vs. NIRS prediction. Based on the lab-measured leaf macronutrients (N, P, K, S, Ca, Mg), NutriLOGIC categorised 45.3%, 33.3% and 21.3% of the sample macronutrient measurements as high, low, and normal, respectively. Whereas, using values predicted by fresh leaf models (this study), NutiLOGIC categorised 43.3%, 29.3% and 27.3% of the sample macronutrient measurements as high, low, and normal respectively (Table S6). The overall error rate of nutrient status classification is 12.7%.

For micronutrients (Na, Zn, Fe, Cu, Mn, B), NutriLOGIC categorised 32.7%, 42.0% and 25.3% of the lab-based micronutrient measurements as high, low, and normal respectively. Using values predicted with fresh leaf models, NutiLOGIC provided a rating of 31.3%, 44.7% and 24.0% of the sample micronutrient measurement as high, low, and normal, respectively (Table S6). The overall error rate of nutrient status classification is 13.3%.

Overall, using NIRS prediction of fresh leaves as compared to laboratory analysis resulted in a nutrient status classification error rate of 12.7% for macronutrients and 13.3% for micronutrients.

Discussion

Macronutrients vs. micronutrients

Generally, macronutrients were predicted at higher accuracies compared to micronutrients via NIRS. These results agree with past studies (Petisco et al., 2005, 2008; Huang et al., 2008; Liao et al., 2012), which found that macronutrients on leaves such as N, P, K, Ca and Mg are generally predicted better compared to micronutrients such as Fe, Zn, Mn and B. Despite this, the prediction accuracies for estimating micronutrients in this study were better compared to other studies that found R2 of 0.00–0.69 for most micronutrients except for Zn (Petisco et al., 2008; Liao et al., 2012; Van Maarschalkerweerd et al., 2013).

Macronutrients were estimated with high accuracy by NIRS in this study because most macronutrients (i.e. N, P and S) exist dominantly in organic form (Droux, 2004; Marschner, 2011). Nutrients in organic form can be directly measured by NIRS, as organic compounds contain chemical bonds such as C–H, N–H, S–H, C–C and C=C that have a unique signature in the NIR range (Richardson, Reeves & Gregoire, 2004). For example, wavelengths used by the model for predicting total N (Figs. 4 and 5) corresponded to protein and chlorophyll compounds, which is also reported by other studies (Al-Abbas et al., 1974; Johnson, 2001; Min et al., 2006; Shao & He, 2013). The distribution of important wavelengths used by the Cubist model in predicting leaf total N is shown in Fig. 5A.

Figure 4 Important wavelengths used in the NIRS to predict total N content of cotton leaves.

The important wavelengths used in the NIR model are located in the 2,225 nm and other important wavelength bands are located in region between 2,200 and 2,500 nm. The level of importance represents the top three most important wavelengths used in the 50 iterations of the Cubist model to predict the leaf nutrients. The construction of the graph is explained in “Statistical Analysis”.

Figure 5 Comparison of important wavelenghts used by the NIRS models to predict total N and Cu.

(A) Important wavelength bands used in the NIRS, dried and ground leaf models to estimate the total N content of cotton leaf. (B) Important wavelength bands used in the NIRS, dried and ground leaf models to estimate the total Cu content of cotton leaf. The main wavelength bands used for estimating total N and Cu are located in the NIR range (2,200–2,400 nm). The level of importance shows the top three most important wavelengths used in the 50 iterations of the Cubist model to predict the leaf nutrients. The construction of the graph is explained in “Statistical Analysis”.

The importance of the reflectance spectra at wavelength 2,225 nm (Fig. 4) indicated the relevance of protein in determining the N content of leaves, as protein corresponds to wavelengths 2,054 and 2,712 nm due to the C–H and N–H bonds (Kokaly, 2001; Bojović & Marković, 2009). These wavelength bands are also correlated to the C–H bonds in the phytol tails of chlorophyll (Sims & Gamon, 2002). Similar results were reported by Min et al. (2006) who found a strong relationship between the N content of Chinese cabbage with reflectance at 2,229 and 2,283 nm.

Despite numerous studies such as Farabee (2003) and Sims & Gamon (2002) showing that chlorophyll, which is more prominent in the visible spectrum due to its C–C and C=C bonds in the porphyrin ring provides an important indication of N content in leave tissues, NIRS was able to accurately total N content. This shows that protein is another important indicator of total N content of leaves.

Another successful example of how macronutrients are predicted by NIRS is the accurate prediction of P content, which was expected because, similar to N, P exists mostly in organic form in plants in the form of phytates (50–70%), nucleic acids, phosphoproteins, phospholipids (20–30%), and the remaining existing as inorganic P (Chen et al., 2002; Petisco et al., 2005). The important wavelengths included 2,390–2,400 nm in the NIR calibration. These wavelength bands were comparable to the findings of De Boever, Eeckhout & Boucque (1994) who found a high correlation between the 2,200–2,400 nm and the total P and phytate P content in vegetables and a slightly lower correlation at the 2,048 nm wavelength. Similarly, Petisco et al. (2005) found that important wavelengths used in predicting P are close to 2,330 nm which are correlated to phospholipids.

Conversely, most micronutrients and several macronutrients such as Mg and Ca exist in inorganic forms that are associated with organic compounds, largely as enzyme cofactors, whereas K is the only nutrient that is not a constituent of any organic compounds (Marschner, 2011). In this study, these macronutrients and micronutrients were well predicted by NIRS. The inability of NIRS to directly detect inorganic compounds suggests alternative mechanisms:

via the association of nutrients with the functional groups of organic compounds or directly to the organic matrix (Huang et al., 2008; Yarce & Rojas, 2012), and

indirectly by measuring organic compounds that can directly be detected by NIRS that are correlated with the nutrients (Sims & Gamon, 2002).

For example, K is an essential macronutrient due to its importance in managing more than 50 enzymes essential in maintaining the functioning of plants (Ciavarella, Batten & Blakeney, 1998; Galvez-Sola et al., 2015). Different from other macronutrients, K is not incorporated into the chemical structure of plant tissues and it is most influential in K+ ionic form or its inorganic form (Ciavarella, Batten & Blakeney, 1998; Prajapati & Modi, 2012) which is not readily detected by NIRS. The wavelength regions used in the NIRS model for K corresponded to wavelengths associated with sucrose between 1,800 and 2,100 nm (Giangiacomo, 2006). This also coincided with the regions associated with other carbohydrates (Rébufa, Pany & Bombarda, 2018).

The association of K with carbohydrates and organic acids and its correlation with the near infrared spectra can be explained by the two alternate mechanisms mentioned above. First, K is influential in photosynthesis and translocation of assimilates such as sucrose, cellulose, and starch; hence, the abundance of these assimilates and organic acids (e.g. malic acids) which can be directly measured by NIRS, can be correlated indirectly to the abundance of K in the plants (Ciavarella, Batten & Blakeney, 1998). Secondly, K also forms cation-carbohydrate complexes that are detectable by NIRS (Cadet & Offmann, 1996). These two mechanisms show that there is a possibility for nutrients that mostly exist in inorganic form in plant tissues to be detected via NIRS.

Another factor that is responsible for the higher NIRS model prediction accuracies of macronutrients is their higher concentration (Clark, Mayland & Lamb, 1987; Liao et al., 2012). In general, the result also agrees with Liao et al. (2012) which stated that reliable NIR calibration could only be achieved with nutrients of concentrations above 100 mg/kg as shown by the poor validation R2 for Zn (0.36) for which its concentration ranged from 8.14 to 64.62 mg/kg. Thus, the comparable validation accuracies of several micronutrients such as Fe, Mn, Na and Mo to macronutrients such as K and P can be explained by their concentrations that are generally higher than 100 mg/kg in terms of abundance (Table 2). Despite this, Cu (<15 mg/kg) which had a lower concentration compared to Zn had a reasonable validation R2 of 0.65. This result contradicts the findings of De Aldana et al. (1995) that found Vis-NIR was unsuccessful in predicting Cu in different plants.

This successful prediction of Cu can be related to its positive correlation to the leaf total N (linear correlation coefficient, r = 0.70), whereas Zn content was not correlated to other well-estimated nutrients. The correlation between Cu and total N explains the validation R2 of Cu (0.65), accounting for approximately 70% of the validation R2 of the total N calibrations (0.94). To infer the mechanism behind the prediction of Cu by NIRS, Fig. 5 shows the important wavelengths used in the NIRS model to estimate Cu and total N.

The wavelength bands used in the NIRS calibrations for estimating Cu coincided with the wavelength bands used to estimate leaf total N (Fig. 5). Marschner (2011) stated that 98% of Cu in plants occur in complex forms with proteins due to its affinity to peptide. Additionally, 50% of Cu in plants exist in the chloroplast taking part in the photosynthesis process (Marschner, 2011). These two factors provide comprehensive support to the high correlation found between the Cu content and leaf total N content.

In summary, NIRS were able to accurately measure macronutrients such as N, P and S directly, as these nutrients are major constituents of NIRS sensitive-organic compounds such as proteins and nucleic acids. Moreover, NIRS can also accurately predict macronutrients and micronutrients that are dominantly present in inorganic form through their association with NIRS sensitive-organic compounds, or as organic complexes, and indirectly through correlated organic compounds. Despite the lower concentrations of micronutrients such as Cu, their association with organic compounds (i.e. proteins) enables it to be measured by NIRS.

Nitrate-N behaves closer to micronutrients in that it exists in a relatively low concentration compared to organic N, on average 3% of total N in this study, and it is categorised as an inorganic compound. Despite the higher concentration of nitrate-N compared to most micronutrients, nitrate-N was poorly predicted by the NIRS model. The inability to predict nitrate-N can be due to the mobility of nitrate-N in plant tissues (Marschner, 2011). Nitrate-N is accumulated in the vacuole (5–75 µM) of plant cells with a small portion existing in the cytosol (1–5 µM); however, nitrate-N is readily transported throughout the plant and it is quickly depleted in certain conditions (Marschner, 2011).

Fresh vs. dried samples

Expectedly, models using dried and ground leaf gave a more accurate estimation of both macro- and micronutrient content of cotton leaves compared to models calibrated using fresh leaf samples. This result coincided with Rotbart et al. (2013) who found that dried and ground olive leaves provide a better prediction accuracy in predicting leaf nutrient content compared to fresh leaf derived models. The major advantage of dried and ground leaf samples is the low moisture content in samples. Water molecules have a strong effect on NIR reflectance spectra, especially at wavelengths of 1,400 and 1,950 nm (Fig. 6) (Rambla, Garrigues & De La Guardia, 1997; Jie et al., 2014). This water influence can be attributed to the vibration of the O-H bond in water molecules (Ludwig & Khanna, 2001).

Figure 6 NIR reflectance spectra of fresh and dried, and ground cotton leaf samples.

Moisture affected regions (1,400 and 1,950 nm) are highlighted. Fresh leaves have a lower NIR reflectance spectrum.

Moreover, dried and ground samples are more homogenous and provide a more complete representation of the sample’s chemical composition. Despite the advantages of using standardised samples, half of the macronutrients assessed (i.e. P, K and Mg) were predicted equally well by fresh leaf models. Macronutrients that were significantly more accurately predicted by dried and ground leaf models (i.e. Total N, Ca and S) had validation RMSEs that were still acceptable when fresh samples were used. Wavelengths used by dried and ground leaf and fresh leaf models in estimating leaf nutrient contents were quite different with some overlap (Fig. 7). Fresh leaf models used a wider range of wavelengths, whereas dried and ground leaf models used a narrower range (Fig. 7). This could be a method for NIRS to compensate for the reduced information in the fresh leaf spectrum from moisture influence.

Figure 7 Comparison of important wavelenght used to determine the total N content of cotton leaves using NIRS models developed from dried and ground, and fresh and removed leaves.

(A) Important wavelength used to estimate total nitrogen content of dried and ground leaf samples. (B) Important wavelength used to estimate total nitrogen content of fresh and removed leaf samples. The dried and ground leaf models used wavelengths at 2,200–2,450 nm, where the fresh and removed leaf model used wavelengths at 1,425–2,500 nm. The level of importance shows the top three most important wavelengths used in the 50 iterations of the Cubist model to predict the leaf nutrients. The construction of the graph is explained in “Statistical Analysis”.

Conversely, micronutrients were not well predicted by fresh leaf models, except for Mn and Cl. The lower prediction accuracy may be attributed to the lower signal of micronutrients, which may be overwhelmed by the water influence in fresh leaf samples. However, the validation RMSEs of the fresh leaf calibration were still acceptable compared to the validation RMSEs of the dried and ground validation RMSEs. On average, using models of fresh and removed leaves instead of dried and ground leaves reduced the prediction accuracy by 19% for macronutrients and 11% for micronutrients. This accuracy loss equates to a 0.04% lower accuracy for total N, and a loss of 7 mg/kg accuracy for Fe, which can be considered acceptable for agronomic purposes. Hence, NIRS has a large potential to be used for in-field application of cotton plant macronutrient analysis. For micronutrients that have low concentration, NIRS can be used as screening tests to indicate deficiencies.

From the two methods for obtaining fresh cotton leaf NIR scan, fresh and intact and fresh and removed leaves were quite similar in accuracy based on the validation RMSE. Several nutrients were better predicted under fresh and intact leaf conditions. However, the overall difference is negligible. The only difference between the two fresh leaves was the moisture content, as removed fresh leaves had, on average, 7% less moisture compared to intact leaves. Moisture content decreased with time as leaf samples were removed from the plant.

This study on fresh and intact leaves was conducted on a glasshouse with a controlled environment. In terms of practicality, acquiring NIR spectra data from fresh and removed leaves is the preferable option, as sample conditions are more controlled, and it avoids extreme terrains or weather conditions. Temperature variation can be an issue for the application of NIRS in-field because it can also affect the NIR spectrum (Roger, Chauchard & Bellon-Maurel, 2003; Nicolai et al., 2007). Temperature can alter the chemical composition of plant tissues, as it affects the abundance of organic compounds (Roger, Chauchard & Bellon-Maurel, 2003; Nicolai et al., 2007). The application of NIRS at different sampling temperatures or moisture conditions can result in inconsistent readings (Roger, Chauchard & Bellon-Maurel, 2003; Minasny et al., 2011). Thus, conducting measurements in a more controlled temperature would be more beneficial.

In the Australian cotton industry, NutriLOGIC is an important decision support tool used by cotton farmers. Currently, due to the delay in acquiring nutrient status information, the use of NutriLOGIC is still inefficient. From the comparisons between lab-measured nutrient status and NIRS predicted nutrients status (using the fresh leaf model), it was found that the error of nutrient classification was 12.7% for macronutrient and 13.3% for micronutrient. The small difference in classification error between macronutrients and micronutrients despite the lower accuracy of micronutrient NIRS models shows that the accuracy of micronutrient NIRS models are acceptable for the purpose of NutriLOGIC. Moreover, this error rate can be compensated by conducting scans on more samples. As NIRS readings are rapid, multiple scans can be done quickly to further reduce error in sampling and readings. This demonstrates that NIRS can accurately obtain plant nutrient status information in real-time, which will improve the efficiency and effectiveness of a decision support system.

Conclusions

In conclusion, NIRS can be a tool that can provide rapid and real-time cotton plant nutrient analysis. All macronutrients, that is total N, P, K, Ca, Mg and S were estimated with high accuracies, and most micronutrients, that is Fe, Mn, Cu, Mo, B, Cl and Na were estimated with relatively high accuracy by NIRS. Macronutrients were generally predicted with higher accuracy by NIRS due to their higher concentration, and important macronutrients such as total N, P and S exist in NIRS sensitive-organic compounds, whereas Ca, Mg, K and micronutrients that exist dominantly in inorganic form were measured through their association with organic compounds or indirectly through correlation with organic compounds.

The small reduction in accuracy from models calibrated using standard dried and ground leaves to the models calibrated using fresh and removed leaves shows that there is a strong potential for NIRS to be used in-field. Moreover, nutrients levels predicted using fresh leaf models are accurate enough to be applied to decision support tools such as NutriLOGIC. This enables rapid nutrient analysis that can prompt effective nutrient management. Further research should directly link NIRS and nutrient management by developing models that use the estimated plant nutrient status information to directly calculate a fertiliser prescription rate based on plant’s growth stage.

Supplemental Information

Supplemental Information 1 Raw data of the nutrient (macronutrient and micronutrient) content values of the cotton leaves samples obtained from chemical data analysis.

Each sample is represented by a sample ID.

Click here for additional data file.

Supplemental Information 2 Image of the cottn leaf samples at different sample preparation.

(A) Fresh and intact cotton leaves scanned using the PhoneLabTM spectrometer. (B) Fresh and removed cotton leaves 1 hour after removal from plant prepared for scanning on aluminium foil background. (C) Dried & ground cotton leaves sieved through a 0.25 mm sieve prepared for scanning on aluminium foil background.

Click here for additional data file.

Supplemental Information 3 Summary of the statistical distribution of the R2 values of the models calibrated using the NIR spectra in predicting the calibration and validation dataset portrayed in Figure 3.

The statistival distribution summary is divided into macronutrient and micronutrient. The summary includes the minimum, 25th interval, mean, median, 75th interval, and maximum R2 value.

Click here for additional data file.

Supplemental Information 4 Summary of the validation accuracy parameters of dried & ground cotton leaf’s micronutrient calibrations constructed with the NIRS spectral range.

The accuracy parameters include the R2, Lin’s concordance, root mean square error (RMSE), and bias. Values are mean of 50 realisations of random data split (n = 375), 75:25 calibration: validation.

Click here for additional data file.

Supplemental Information 5 Summary of the validation accuracy parameters of dried & ground cotton leaf’s micronutrient calibrations constructed with the NIRS spectral range.

The accuracy parameters include the R2, Lin’s concordance, root mean square error (RMSE), and bias. Values are mean of 50 realisations of random data split (n = 375), 75:25 calibration: validation.

Click here for additional data file.

Supplemental Information 6 Summary of the validation accuracy parameters of the dried and ground, and fresh and removed cotton leaf’s macronutrient and micronutrient models calibrated from the NIR spectra (n = 229).

The validation accuracy parameters includes the R2, Lin’s concordance, root mean square error (RMSE), and bias which are the mean of 50 realisations of random data split, 75:25 calibration: validation.

Click here for additional data file.

Supplemental Information 7 Summary of the validation accuracy parameters of the fresh and removed, and fresh and intact cotton leaf’s macronutrient and micronutrient models calibrated from the NIR spectra (n = 115).

The validation accuracy parameters include the R2, Lin’s concordance, root mean square error (RMSE), and bias which are the mean of 50 realisations of random data split, 75:25 calibration: validation.

Click here for additional data file.

Supplemental Information 8 Confusion matrix which shows the effectiveness of fresh leaf NIRS models in the application of nutrient decision support tools such as NutriLOGIC (nutrient decision support system).

The matrix identifies deficient or excessive levels of various macro- and micronutrients in a 25 random cotton leave samples from the Australian Cotton Research Institute (ACRI). The elements were compiled into macronutrients (N, P, K, S, Ca, Mg) and micronutrients (Na, Zn, Fe, Cu, Mn, B). The classification error rates were calculated by subtracting the measured percentage by the predicted percentage.

Click here for additional data file.

Supplemental Information 9 Reasoning why Cubist regression method was selected for calibration instead of PLSR method.

The reasoning includes some prove that the cubist method has shown to be superior compared to PLSR in this study and other studies.

Click here for additional data file.

The authors thank Ms. Kellie Gordon and Ms. Phebe Samrani for their technical assistance. BM acknowledges the contribution from Sydney Institute of Agriculture’s Nitrogen use efficiency project.

Additional Information and Declarations

Competing Interests

Author Contributions

Data Availability

Timothy Weaver is an employee of CSIRO.

Jeremy A. Prananto and Budiman Minasny declare that they have no competing interests.

Jeremy Aditya Prananto conceived and designed the experiments, performed the experiments, analyzed the data, prepared figures and/or tables, authored or reviewed drafts of the paper, and approved the final draft.

Budiman Minasny conceived and designed the experiments, analyzed the data, authored or reviewed drafts of the paper, and approved the final draft.

Timothy Weaver conceived and designed the experiments, authored or reviewed drafts of the paper, and approved the final draft.

The following information was supplied regarding data availability:

The raw chemical data analysis of the cotton leave samples is available in the Supplemental Files.

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
