# Peer review of "Rapid and cost-effective nutrient content analysis of cotton leaves using near-infrared spectroscopy (NIRS)"

_PeerJ, doi:10.7717/peerj.11042_

## Round 0.1 · original submission · Major Revisions

The manuscript has been reviewed by two reviewers and both found it an interesting study. However, they also had some issues on its relevant literature about NIRS and Cubist model and also on its readability and legitimacy of figure legends. I suggest authors to revise the manuscript accordingly and resubmit.

Reviewer 1 ·

Basic reporting

The manuscript introduced a nice study to use NIR as a rapid tool for the analysis of macronutrients and micronutrients contents of cotton leaves. The experiment was performed on the leaf samples of three states, Fresh Intact, Fresh Removed, and Dried Ground. The study compared the predictive accuracy of NIR between macronutrients and micronutrients, between fresh and dried leaves. The manuscript was well prepared and easy to follow. NIR analysis of plant leaves for nutrients was done on dried and fresh samples. This study did both and compared the results, which would be adding new information for the research community. The discussion on the ability of NIR to predict low-concentration micronutrients in fresh state was relevant and insightful. A few comments for the authors to improve.
Line 109-111. Recently there are papers on NIRS analysis of plant leafs of field crops such as corn and wheat. Encourage the authors to search for the recent literature.
Table 1 is not easy to understand. Your description in the text was clear on the numbers of samples from each scanning state. Consider removing Table 1 to avoid redundancy and confusion.
Figures 4, 5 and 7 are not clear to me. First, I cannot understand what “wavelength importance” was and how it was derived. Second, I did not understand “Level of Importance”, First, Second and Third. Could you please add more information in this regard?
Table 3, if I understand correctly, the fresh and removed cotton leafs had 114 samples and dried cotton leafs had 114+146 samples. Did this comparison run only on the 114 common samples for both sets? Because if not, then the result of favorance to “Dried and Ground” for most leaf attributes could be due to the difference in the datasets rather than the scanning state?

Experimental design

Experimental design is appropriate.

Validity of the findings

Findings are valid and supported by the data presented.

Additional comments

Very good work. Please see my basic reporting for improvement.

·

Basic reporting

Thank you to the authors for the manuscript titled “Rapid and cost-effective nutrient content analysis of cotton leaves using near-infrared spectroscopy”. Overall, I believe that the project conducted is important and it would be helpful to many readers. This is a good, self-contained study that will be helpful for many readers who intend to use near infrared spectroscopy. The study provides a good overview of the literature, the discussion was well-written and the authors have provided the raw data and supplementary information that will also be helpful to readers.

Most of my comments relate to the readability of the manuscript and highlight where some clarity is needed. I suggest that the overall manuscript will need to be heavily edited to improve overall readability of the manuscript and the figures legends. Authors will need to fix grammatical errors and restructure some of the paragraphs. Specifically, many of my comments come from the results section. While the results are informative, I believe that the results do not flow together as there are many “paragraphs” with only one sentence. The manuscript would be greatly improved if authors spend some more time rewriting this section. Furthermore, while the authors reported the R2 for various macro- and micronutrient models, there was no mention of the RMSE for each of the models. Please add the RMSE values for each model as it is important for readers to know the amount of error (RMSE) associated with predicting various traits.

Experimental design

This original primary research manuscript appears to follow the Aims and Scope criteria of PeerJ. The overall research question was well-defined and provides evidence for why this project is needed, both from a scientific and industry view point. Furthermore, it is an interesting paper as the authors have used a less common technique, Cubist regression models, for developing the NIR models. There are some issues regarding the clarity of their methods which I have highlighted in "General comments for the author".

Validity of the findings

The findings of the manuscript seem reasonable given the samples the used and the plant traits they measured. Further, the manuscript provides information on potential issues that can arise when moving from laboratory-based measurements to field-based measurements. This is helpful for readers across many different research fields who wish to use NIR.

Additional comments

I have provided some comments below regarding the readability of the manuscript.

Line 29 – 30. Please edit R2 to either r2 or R2.
Line 35. I am not sure what you mean by the word “effective”. Please consider using a different word or provide some clarification in how it is more effective.
Line 78 – 81. Please consider editing or provide further clarification in the sentence. I am not sure how unreliable measurements of plant available nutrients can be an issue for soil analysis. Is it a separate issue or is there a link that is missing in the sentence?
Line 77 – 88. Overall this paragraph will need some editing from the authors. It is not clear if the methods and issues described relate to measurements on soil or plant. Perhaps add some details to clearly state soil analyses look at overall nutrient content in soil and this overall nutrient content does not necessarily equal the amount of nutrients available for a plant. To make this point clearer, please provide an example(s) of this and include a reference(s). The He et al. 2007 doesn’t seem to best reference for this paragraph. While it does indicate that one can measure nutrients in soil using NIRS technology, it doesn’t highlight that soil nutrient content doesn’t equal nutrient availability to plants.
Line 100 – 102. Consider editing this sentence to make it more concise.
Line 121 – Remove “possibly”.
Line 123 – 124. I am not sure if the last sentence of this paragraph is required.
Line 132. What are nutrient decision support tools, how are they typically used and why is it necessary to link leaf nutrient status information to them?
Line 137/183. Please provide timeline of data collection, from when were leaves collected to when samples were scanned. The sequence of data collection is not clear.
Line 141. Consider inserting paragraph about the spectrometers (Line 195-204) before this paragraph on spectrometer set up and calibration. This would help link set up protocol to actual instrument as this can vary across spectrometer brands.
Line 160. Were the exact same leaves used for the “fresh” scan and then picked for the “fresh and removed” and “dry and ground” scans? If different leaves were used, please include this in your methods and provide a sentence about the comparability of these samples.
Line 169-174. Please revise the order of these sentences and edit for clarity. I assume that petioles were removed prior to grinding and that scanning was performed on the ground leaf samples. Please ensure that the sequence of these sentences reflects this. Line 173 can be interpreted that the petioles were removed then processed and scanned, is this correct or did you analyse the leaf section only? Also, it states “scanned with a similar method to the dried and ground leaf samples” but this section is designated for the dried and ground leaves”. Please clarify.
Line 175. Replace “poured” with spread.
Line 231. Edit to “analysis are shown in Table 2”.
Line 243 – 249. How/why did you choose this particular combination of treatments? Did you try multiple combinations of pretreatments and this performed the best or did you use a specified pretreatment from another project (and if so, please reference).
Line 250. I assume Figure 1 shows the raw NIR spectra and perhaps should be mentioned somewhere in this paragraph.
Line 253. Do you mean “(section 2.3) and the nutrient analysis data (section 2.2)”?
Line 259 – 267. This comment is to highlight that I am not familiar with Cubist regression tree models and so I am unable to provide advice on the set up for this type of analysis. I appreciate that models using partial least squares regression (PLSR) were also developed and compared with the Cubist models. Given that these PLSR models were developed and compared, it would be helpful to provide information on these methods in the supplementary materials. For example, were the same pre-processed spectra used for the two types of models? Also please explain why the Cubist approach was better, for example did you compare R2 and RMSE?
Line 269 – 274. This section of the first sentence makes this paragraph misleading to readers; “the other 25% was used as validation data to assess the predictive accuracy of the model”. In order to assess the predictive accuracy of a model, one wants to see how the model performs on new samples that have not been used to train the model. As soon as validation samples are introduced to the model, they are no longer considered to be independent/validation samples and therefore cannot provide information on the predictive accuracy of the model for new samples. Because of Line 272-73 “The 75:25 data split and calibration processes were repeated 50 times to gain an average...”, I interpret this as the initial validation samples were reintroduced to the train the model at some point during the 50 x 75:25data splits. This resembles cross-validation of the model, rather than predictive accuracy. Please consider rephrasing the paragraph to either a) describe the cross-validation methods and provide reasoning for why there are no validation samples for the model or b) redo the analysis where the 25% validation samples are left aside for prediction tests and never included during model calibration. If option a is chosen, randomly splitting a dataset for cross-validation may not necessarily be the best technique to assess how the model will perform for new samples, it simply tells you how good it can perform for your dataset. To get a better understanding on how it will perform for new samples, it is likely that there is some nested structure in the data set; samples were collected across different days/different plots etc. Splitting up the dataset using nested structure during cross-validation will provide more information on the feasibility of using NIR to assess nutrient content across cotton leaves. For more information on cross-validation and prediction techniques: Au, J. et al. 2020, Sample selection, calibration and validation of models developed from a large dataset of near infrared spectra of tree leaves. Journal of Near Infrared Spectroscopy.
Line 288 – 290. What NIR models were compared and what were the different treatments? Do you mean the different leaf treatments, intact vs dried or do you mean different spectra treatments?
Line 307. Ensure to add a description of a nutrient decision support system and how it is used. This will help the reader better understand the methods in Line 314-319.
Line 323 – 329. To me, this paragraph seems unnecessary but could be useful for other people.
Line 332. What is the “statistical distribution of the average R2 values of the…”
Line 335. What does “used as standards” mean? Is that in reference for comparing the different treatments?
Line 332 – 338. This appears to describe methodology and I don’t believe it was included in the methods section. Please consider rephrasing or moving it to the methods section.
Line 342 – 346. It is not clear if there was only one validation set or multiple validations used in Figure 3.
Line 381 – 383. This looks like it should be in the discussion.
Line 417 - 419. Why isn’t this critical? Should it be introduced in the discussion rather than the results?
Line 424 – 426. This sentence is confusing, please consider rephrasing it.
Line 430 – 434. The numbers appear to be incorrect as 32.7% = high, whereas 42.0% = low. Also, how is the error rate of nutrient classification performed? Was this written in the methods?
Line 436 – 439. I am not sure how “this shows that the error of the models is within an acceptable margin”.
Line 450. If this is an example for the previous paragraph, please keep the sentences together.
Line 533 – 534. Please add this floating sentence to a paragraph.
Line 592 – 593. If known, please provide a reference to support that the prediction of micronutrients is negatively affected by the water in fresh leaves.
Line 665. Please ensure that all references are checked and all follow a consistent format.
Figure 2. It appears that the same example of spectra is shown in Figure 1 and 2. I suggest that this is not necessary and to remove Figure 2b.
Figure 6. It is not clear what “a lower NIR reflectance spectrum” means.
Table 1. If a left-aside validation set is used (i.e. one that is not used for calibration) please add information in this table to describe how many samples were used for calibration vs. validation.

---

## Round 0.2 · Minor Revisions

Thank you for revising the manuscript. There are two minor things that I suggest authors to consider addressing:

1) Line: 126-128. Limit the cited references to 4 to 5 at maximum, and mention specifically what they found, if you can.
"tThere are few published results in the literature that have explored the use of NIRS for plant nutrient analysis (Au et al., 2020; Min et al., 2006; Menesatti et al., 2010; Liao et al., 2012; Martínez-Valdivieso et al., 2019; Savi et al., 2019; Zerner and Parker, 2019; Zhang et al., 2019; Cuq et al., 2020; Dedeoglu, 2020; Johnson et al., 2020)."

2) Figure legends are too small to read, make it legitimate (Figure 4, Figure 5, and Figure 7).

---

## Round 0.3 · accepted · Accept

Thanks for revising the manuscript according to reviewer's and editor's comments. Your response has been considered positively, and the manuscript is now accepted for publication. Congratulations.